# DRONE: Data-aware Low-rank Compression for Large NLP Models

**Patrick H. Chen**
UCLA
Los Angels, CA
patrickchen@g.ucla.edu

**Hsian-fu, Yu**
Amazon
Palo Alto, CA
rofu.yu@gmail.com

**Inderjit S. Dhillon**
UT Austin & Amazon
Palo Alto, CA
inderjit@cs.utexas.edu

**Cho-jui, Hsieh**
UCLA & Amazon
Los Angels, CA
chohsieh@cs.ucla.edu

## Abstract

The representations learned by large-scale NLP models such as BERT have been widely used in various tasks. However, the increasing model size of the pre-trained models also brings efficiency challenges, including inference speed and model size when deploying models on mobile devices. Specifically, most operations in BERT consist of matrix multiplications. These matrices are not low-rank and thus canonical matrix decompositions do not lead to efficient approximations. In this paper, we observe that the learned representation of each layer lies in a low-dimensional space. Based on this observation, we propose DRONE (**d**ata-awa**r**e l**o**w-ra**n**k compr**e**ssion), a provably optimal low-rank decomposition of weight matrices, which has a simple closed form solution that can be efficiently computed. DRONE can be applied to both fully-connected and self-attention layers appearing in the BERT model. In addition to compressing standard models, our method can also be used on distilled BERT models to further improve the compression rate. Experimental results show that DRONE is able to improve both model size and inference speed with limited loss in accuracy. Specifically, DRONE alone achieves 1.92x speedup on the MRPC task with only 1.5% loss in accuracy, and when DRONE is combined with distillation, it further achieves over 12.3x speedup on various natural language inference tasks.

## 1   Introduction

The representations learned by large-scale Natural Language Processing (NLP) models such as BERT and its variations have been widely used in various tasks [8, 2, 25, 3, 24]. The successes of these large NLP models rely on the usage of large corpus and big models. Indeed, researchers have reported better results with models that have more parameters [31] and number of layers [1]. The increasing model size of the pre-trained models inhibits public users from training a model from scratch, and it also brings forth efficiency challenges, including inference speed and model size when deploying models on mobile devices.

To deal with efficiency issues, most existing work resorts to adjusting the model structure or distillation. For instance, [17] used locality-sensitive hashing to accelerate dot-product attention, [18] used repeating model parameters to reduce the size and [44] applied a pre-defined attention pattern to save computation. A large body of prior work focused on variants of distillation has also been explored

35th Conference on Neural Information Processing Systems (NeurIPS 2021).

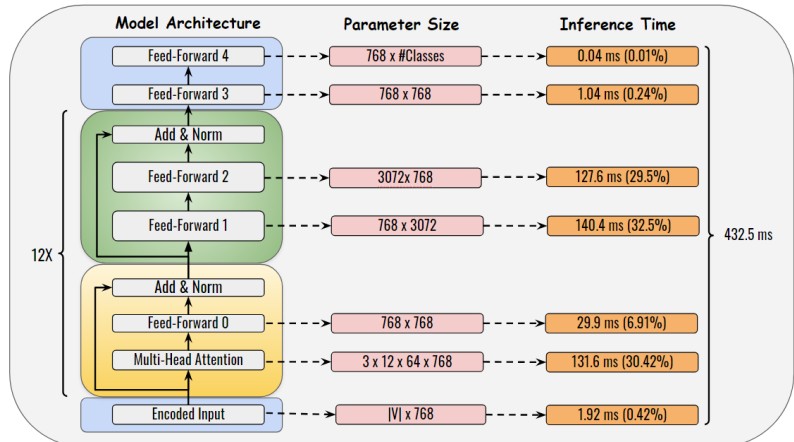

Figure 1: Illustration of the BERT-base computational model. $|V|$ (at bottom of the Parameter Size column) denotes the number of tokens in the model. #Classes (at top of the Parmeter Size column) denotes the number of classes in the down-stream classification task. Input encoding, Feed-forward 3 and Feed-forward 4 are computed only once and thus do not contribute much to overall time. The inference time (in milliseconds) listed here is based on the inference time measured on a CPU.

[27, 15, 35, 20, 34, 35, 41, 45, 4]. These methods require a specific design of model architecture, or a long training stage and thus it is less straightforward to combine these methods with each other.

In this paper, we explore a simpler acceleration method to speed up inference time which can be applied to most existing architectures. As shown in Figure 1, matrix multiplication (feed-forward layer) is a fundamental operation which appears many times in the Transformer [36], the backbone architecture of the BERT model. In fact, the underlying computation of both multi-head attention layers and feed-forward layers is matrix multiplication. Therefore, instead of resorting to the complex architecture redesign approaches, we aim to investigate whether low-rank matrix approximation, a classical and simple model compression approach, can be used to accelerate Transformers. Despite its successful application to CNNs [42, 33, 32], at first glance, low-rank compression does not appear to work for BERT since the matrices in both feed-forward layers and attention layers **are not low rank** (see Figure 2). Therefore, even the optimal low-rank approximation (e.g., by SVD) will lead to very large reconstruction error. This is probably why low-rank approximation has not been successfully used in BERT compression.

In this paper, we propose a novel low-rank approximation algorithm to compress the weight matrices even though they are not low-rank. The main idea is to **exploit the data distribution**. In NLP applications, the latent features (features fed into each matrix mulitplication layer) usually indicate some information extracted from natural sentences, and they often lie in a subspace with a low intrinsic dimension [5, 32, 22]. Therefore, in most of the matrix-vector products, even though the weight matrices are not low-rank, the input vectors lie in a low-dimensional subspace, allowing dimension reduction with minimal degraded performance. We mathematically formulate this generalized low-rank approximation problem which includes the data distribution term and provide a closed-form solution for the optimal rank-$k$ decomposition of the weight matrices. By leveraging the data distribution idea, we propose DRONE (**d**ata-awa**r**e l**o**w-ra**n**k comp**r**ession). Our decomposition significantly outperforms the SVD under the same rank constraint, and can successfully accelerate the BERT model without sacrificing too much test performance. In addition to compressing standard models, DRONE can also be used on distilled BERT models to further improve the compression rate. For example, DRONE alone achieves 1.92x speedup on the MRPC task with only 1.5% loss in accuracy, and when combined with distillation, DRONE achieves over 12.3x speedup on various natural language inference tasks.

## 2   Related Work

Fast inference is important for deploying NLP models in various applications. Generally speaking, inference efficiency can be enhanced by hardware [30] or lower-level instruction optimization [23]. On the other hand, the main focus of the current research is on using algorithmic methods to reduce computational complexity. These methods can be mainly categorized into two aspects: attention complexity reduction and model size reduction.

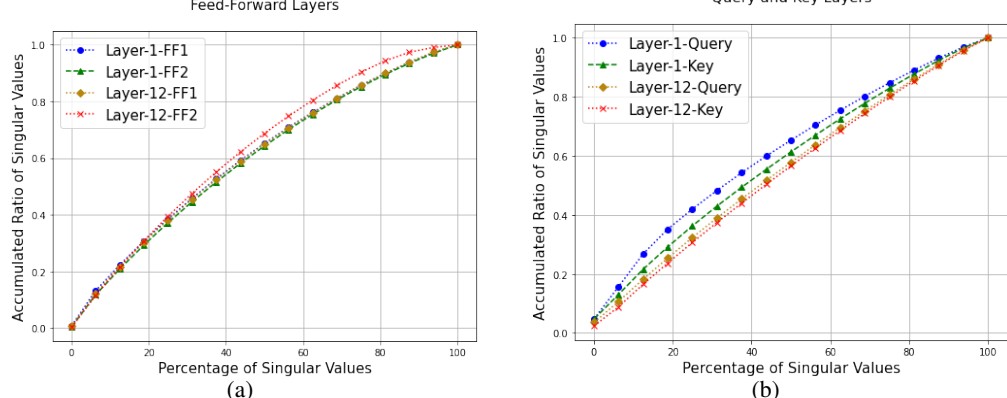

Figure 2: Illustration of the empirical observation that weight matrices in BERT model are not low-rank. The X-axis represents what percentage of singular values; the Y-axis represents sum of singular values connected to the selected ranks divided by sum of all singular values. Ideally, a low-rank structure will have a larger area under the curve, meaning that a small percentage of the singular values can explain their total sum. We observe that the sum of the top 50% of the ranks only accounts about 60% of all singular values for matrices in the BERT model. This shows that the matrices do not have a clear low-rank structure.

**Attention Complexity Reduction**

Attention mechanism is the building block of transformer models and has attracted the most attention of researchers recently in the NLP field [36]. Pre-training on large corpus of BERT, a transformer-based model, has contributed to state-of-the-art performance on various tasks after fine-tuning [8]. Attention on sequences of length $L$ is $O(L^2)$ in both computational and memory complexity, which yields long inference time when the sequence is long. Thus, researchers have focused on reducing the complexity of the attention module. [17] used locality-sensitive hashing to reduce the complexity to $O(L \log L)$. [44, 7] pre-defined an attention map to have a constant computational time. [11] progressively eliminated the redundant context vectors within the attended sequence to improve efficiency of attention in the last few layers of the model. [38] proposed to train the low-rank attention by choosing a rank $r \ll L$. This is similar to our work in the sense of leveraging low-rank structures. But our method does not require training the model from scratch and can be applied to different modules other than attention. In fact, most of the above methods require special modules and thus need to train the proposed models from scratch. This prohibits the usage of a large body of publicly available open models for faster research progress. More importantly, these methods mainly focus on the long sequence scenario. As shown in Figure 1, we have found out that attention module is actually not the main inference bottleneck of inference time in common usage. In most, if not all, models of common usages, two layers of large feed-forward layer are appended after the attention module which incurs much more computational time. Attention complexity reduction only works when a long sequence is used but in current practice this is unusual. Thus, in many tasks accelerating the attention module itself does not contribute to a significant reduction of overall inference time.

**Model Size Reduction**

Inference speed is also related to model compression. In principle, smaller models lead to reduction in the number of operations and thus faster inference time. [28] explored pruning methods on BERT models to eliminate redundant links, and there is a line of research on pruning methods [12, 13, 6, 10]. Quantization methods [43, 14, 19, 9] convert the 32 bits float models into fewer-bits fixed-point representation and make model prediction faster with fixed point accelerator. [18] reduce the model size by sharing encoder parameters. A large body of prior work focused on variants of knowledge distillation [27, 15, 35, 20, 34, 35, 41, 45, 4]. These methods use different strategies to distill information from a teacher network and reduce the number of layers [27] or hidden dimension size [15]. Further, a hybrid compression method by combining matrix factorization, pruning and knowledge distillation is proposed by [21]. Notice that [21] performed SVD for some components and in this paper we propose an improvement over SVD by leveraging input distribution to each layer. Idea of using input distribution to compress model has also been explored in PCN method

[37], which is perhaps the closest to our work. However, DRONE differs from PCN in following three aspects. First, PCN only considers input distribution but not weight matrix and thus it's a special case of DRONE (i.e., $W$ be an identity matrix in equation (3)). Second, PCN merely does dimension reduction whereas our formulation achieves dimension reduction and low-rank approximation simultaneously. Last, PCN does not guarantee the obtained transformation is the optimal; whereas, DRONE formulates an approximation optimization problem and we provide the optimal solution. Other forms of low-rank learning strategies including initialization and structure pruning were also explored in the literature [39, 16], and we will compare to these baseline methods. Among the above-mentioned methods, quantization requires hardware accelerator to maximally reduce the inference time. Pruning methods can only reduce the model size, but the inference time might not be reduced due to the limitation of sparse operations. Only algorithmic methods such as distillation serve as a more generic inference time accelerating method. We want to emphasize that our method is orthogonal to these distillation methods. In fact, the proposed method is an acceleration method that is applicable to all components in most NLP models. In Section 4, we show that DRONE can be combined with the distilled models to further improve the performance.

## 3 Proposed Method

We now introduce an algorithm for improving efficiency of matrix multiplication. The computation of feed-forward (FF) layer in the attention models can be described as:

$$h = Wx + b, \tag{1}$$

$$o = \sigma(h), \tag{2}$$

where $W \in \mathbb{R}^{d_2 \times d_1}$ and $b \in \mathbb{R}^{d_2}$ are model parameters, $x \in \mathbb{R}^{d_1}$ is the latent representation of a token, and $h \in \mathbb{R}^{d_2}$ is the intermediate representation before the activation function, $\sigma(\cdot)$ is the activation function, and $o \in \mathbb{R}^{d_2}$ is the output. Assuming the sequence length is $L$, all the token representations $x_1, \ldots, x_L \in \mathbb{R}^{d_1}$ will pass through this same operation, so in practice the whole FF layer can be computed by a matrix-matrix product $W[x_1, \ldots x_L] + b$, and the computation of the bias term $b$ would be broadcast to all $L$ input tokens. In practice we will normally have $L \ll \max(d_1, d_2)$ (e.g., $L = 128$, $d_2 = 3072$). Notice that applying $\sigma(\cdot)$ on $h$ element-wisely costs $O(Ld_2)$, which is much smaller than the cost of computing $Wx$ ($O(Ld_2d_1)$). Therefore, in this paper we focus on reducing the cost of computing $Wx$ to accelerate the computation. A standard way to accelerate this computation is to perform low-rank approximation on $W$. A low-rank approximation can be obatined by using singular value decomposition (SVD), which achieves the best rank-$k$ approximation in terms of Frobenius norm and we can write $W$ as:

$$W = USV^T \approx U_{W,k} V_{W,k}{}^T,$$

with orthogonal matrices $U \in \mathbb{R}^{d_2 \times d_2}$, $V \in \mathbb{R}^{d_1 \times d_1}$ and a diagonal matrix $S \in \mathbb{R}^{d_2 \times d_1}$. $U_{W,k} \in \mathbb{R}^{d_2 \times k}$ and $V_{W,k} \in \mathbb{R}^{d_1 \times k}$ are the rank-$k$ approximation matrices by taking $U_{W,k} = US_k^{\frac{1}{2}}$, $V_{W,k} = VS_k^{\frac{1}{2}}$, where $S_k^{\frac{1}{2}}$ is the square-root of the first $k$ entries of the diagonal matrix $S$. Given such an approximation, we can simplify the computation in (1) by

$$h = Wx + b \approx U_{W,k} V_{W,k}{}^T x + b.$$

After conducting rank-$k$ approximation, the computational complexity reduces from $O(d_2d_1)$ to $O((d_1+d_2)k)$. When $k$ is small enough, low-rank approximation not only accelerates the computation [32] but also compresses the model size [26]. However, as shown in Figure 2, matrices in FF layer of BERT do not show obvious low-rank structures. We observe that choosing top 50% rank (e.g., $k = 0.5\min(d_1, d_2)$) can only achieve around 60% of the accumulation ratio of singular values, which implies large matrix approximation error. In the meantime, the complexity is still about $O(d_2d_1)$ and there is no enhancement of speed.

Even though the matrices in the model are not low-rank, we now provide an illustrative example to show that a low-rank computation could still exist when data distribution lies in a lower intrinsic dimension. Suppose we have a matrix $W$ defined as below and the input $x$ lies in a subspace:

$$W = \begin{bmatrix} 7 & 0 & 2 & 3 & 1 \\ 9 & 6 & 7 & 5 & 0 \\ 6 & 1 & 8 & 0 & 3 \\ 4 & 3 & 2 & 1 & 4 \\ 1 & 2 & 2 & 1 & 2 \end{bmatrix}, \quad x \in \text{span}\left( \begin{bmatrix} 2 \\ 2 \\ 5 \\ 5 \\ 4 \end{bmatrix}, \begin{bmatrix} 1 \\ 1 \\ 2 \\ 2 \\ 6 \end{bmatrix} \right).$$

In this case, $W$ is a full-rank matrix so there is no lossless low-rank approximation of $W$. On the other hand, the input data $x$ lies in a 2-dimensional subspace so that we could construct the following low-rank approximation:

$$\underbrace{\begin{bmatrix} 7 & 0 & 2 & 3 & 1 \\ 9 & 6 & 7 & 5 & 0 \\ 6 & 1 & 8 & 0 & 3 \\ 4 & 3 & 2 & 1 & 4 \\ 1 & 2 & 2 & 1 & 2 \end{bmatrix}}_{W} \underbrace{\begin{bmatrix} 2 & 1 \\ 2 & 1 \\ 5 & 2 \\ 5 & 2 \\ 4 & 6 \end{bmatrix} \begin{bmatrix} a \\ b \end{bmatrix}}_{x}$$

$$= \underbrace{\begin{bmatrix} 43 & 23 \\ 90 & 39 \\ 66 & 41 \\ 45 & 37 \\ 29 & 21 \end{bmatrix}}_{U} \underbrace{\begin{bmatrix} -1 & -1 & 0.5 & 0.5 & 0 \\ -0.5 & 0 & 0 & 0 & 0.25 \end{bmatrix}}_{V^T} \underbrace{\begin{bmatrix} 2 & 1 \\ 2 & 1 \\ 5 & 2 \\ 5 & 2 \\ 4 & 6 \end{bmatrix} \begin{bmatrix} a \\ b \end{bmatrix}}_{x},$$

which gives a rank-2 matrix $UV^T$ where $W \neq UV^T$ but $Wx = UV^Tx$ for any $x$ in the low dimensional space. This shows that even if $W$ cannot be approximated, it is still possible to construct a good low-rank decomposition, and the key is to exploit the space of input vectors.

## 3.1 DRONE: Data-aware Low-rank Compression

Assuming the input $x$ of the FF layer follows some distribution, instead of minimizing the approximation error of the weight matrix (for which SVD is optimal), we want to minimize the approximation error of the outputs. Denoting $X$ as the $\mathbb{R}^{d_1 \times n}$ matrix where columns of $X$ capture the empirical distribution of the input (when n is large), our goal is to find projection matrix $V_{X,k} \in \mathbb{R}^{d_1 \times k}$ and recovery matrix $U_{X,k} \in \mathbb{R}^{d_2 \times k}$ such that the output is well approximated. We rewrite (1) as:

$$h = WX + b \approx WU_{x,k}V_{x,k}{}^T X + b$$
$$= (WU_{x,k})V_{x,k}{}^T X + b = W_{X,k}V_{x,k}{}^T X + b,$$

where $W_{X,k} = WU_{x,k}$. Intuitively, when $X$ lies in a lower-dimensional space, we could find such a pair by PCA decomposition on $X$ to project $X$ onto the subspace that explains the most variance. In this way, instead of considering the decomposition of $W$, we leverage the distribution of $X$ to complete the low-rank approximation.

However, the best way is to consider the properties of both $W$ and $X$ simultaneously, and we can mathematically present this desideratum by the following optimization problem:

$$\min_{M} \|WX - WMX\|_F^2, \quad \text{s.t.} \quad \text{rank}(M) = k, \tag{3}$$

where $M$ is the desired rank-$k$ transformation which maximally preserves the results of the matrix multiplication. In the theorem below, we show that there exists a closed-form, optimal solution for the above optimization problem. Before stating the theorem, we first introduce some notation. Assuming rank$(W) = r$ and rank$(X) = t$, we can write $W = U_W S_W V_W^T$ and $X = U_X S_X V_X^T$ such that

$$U_W = \begin{bmatrix} U_{W,r} & \bar{U}_{W,r} \end{bmatrix}, S_W = \begin{bmatrix} S_{W,r} & 0 \\ 0 & 0 \end{bmatrix}, V_W = \begin{bmatrix} V_{W,r} & \bar{V}_{W,r} \end{bmatrix}$$

$$U_X = \begin{bmatrix} U_{X,t} & \bar{U}_{X,t} \end{bmatrix} , S_X = \begin{bmatrix} S_{X,t} & 0 \\ 0 & 0 \end{bmatrix} , V_X = \begin{bmatrix} V_{X,t} & \bar{V}_{X,t} \end{bmatrix}.$$

In other words, the decomposition $U_W S_W V_W^T$ and $U_X S_X V_X^T$ are the full-SVD decompositions of $W$ and $X$, respectively. The matrices $U_{W,r}, V_{W,r}, U_{X,t}, V_{X,t}$ denote corresponding row spaces and column spaces, while $\bar{U}_{W,r}$, $\bar{V}_{W,r}$, $\bar{U}_{X,t}$ and $\bar{V}_{X,t}$ are null spaces. With this notation, we are ready to state the theorem.

**Theorem 1.** *Assume rank$(W) = r$ and rank$(X) = t$. The closed form solution $M^*$ of the optimization problem (3) is*

$$M^* = V_{W,r} S_{W,r}^{-1} Z_k S_{X,t}^{-1} U_{X,t}^T, \tag{4}$$

*where $Z_k$ is the rank-k truncated SVD of $Z = S_{W,r} V_{W,r}^T U_{X,t} S_{X,t}$.*

The proof of Theorem 1 is provided in Appendix A. We note that since $Z_k$ is the rank-$k$ truncated SVD of $Z$, we could also write $Z_k$ as $U_{Z,k}V_{Z,k}^T$ by distributing the top-$k$ singular values of $Z$ into left or right singular matrices. Thus the original computation can be rewritten as:

$$WX \approx (WV_{W,r}S_{W,r}^{-1}U_{Z,k})(V_{Z,k}^T S_{X,t}^{-1}U_{X,t}^T)X = U^* V^{*T}X, \tag{5}$$

where $U^* = WV_{W,r}S_{W,r}^{-1}U_{Z,k}$ and $V^{*T} = V_{Z,k}^T S_{X,t}^{-1}U_{X,t}^T$ are two rank-$k$ matrices, and we will replace $W$ by $U^* V^{*T}$.

## 3.2 Extension to Dot-product Attention

Although the optimization problem in (3) is proposed for feed-forward computation, in this section we show that it can also be applied to the dot-product part of the attention module. The key computation in the attention layer is to compute pairwise similarity between queries and keys of the sequence:

$$O = (Q\bar{Y})^T(KY), \tag{6}$$

where $\bar{Y} \in \mathbb{R}^{d_1 \times n}$ is the batch query data, $Q \in \mathbb{R}^{d_2 \times d_1}$ is the query transformation matrix, $Y \in \mathbb{R}^{d_1 \times m}$ is the batch key data, $K \in \mathbb{R}^{d_2 \times d_1}$ is the key transformation matrix and $n, m$ are query and key batch sizes, respectively. We can again see that the desired low-rank approximation is the solution of the following optimization problem:

$$\min_M \|(Q\bar{Y})^T(KY) - (Q\bar{Y})^T M(KY)\|_F^2, \text{s.t. rank}(M) = k. \tag{7}$$

With $Q\bar{Y} = W$ and $KY = X$, we get the following corollary from Theorem 1 directly.

**Corollary 1.** *Assume rank($Q\bar{Y}$) = $r$ and rank($KY$) = $t$. Let $Q\bar{Y} = U_W S_W V_W^T$ and $KY = U_X S_X V_X^T$ be the SVD decomposition of $Q\bar{Y}$ and $KY$ respectively. The closed form solution $M^*$ of the optimization problem (7) is given by*

$$M^* = V_{W,r}S_{W,r}^{-1}Z_k S_{X,r}^{-1}U_{X,r}^T, \tag{8}$$

*where $Z_k$ is the rank-$k$ truncated SVD of $Z = S_{W,r}V_{W,r}^T U_{X,t}S_{X,t}$ .*

## 3.3 Overall Algorithm

We have shown that the proposed DRONE method is a generic acceleration module applicable to all parts of neural language models. We summarize the DRONE on feed-forward layer in Algorithm 1. Since in practice we don't have the exact distribution of $X$, we use training data to calculate the low-rank approximations as described in Algorithm 1. The attention map calculation can be done by the same procedure with $W = (Q\bar{Y})^T$ and $X = KY$ as given by the Corollary 1.

To accelerate the whole model, we need to select appropriate ranks for each component. However, since the approximation of one component affects the distribution of overall representations, the optimal rank for the model requires a complete search of all possible combinations of rank values, which is infeasible in practice. We thus resort to a simplified approach as shown in Algorithm 2. A more detailed description is provided in the Appendix B. In short, as the changes of lower layer parameters will cause the distribution of representation shifts in upper layers, we approximate each component one-by-one in their topological order of the model. In another words, we approximate the model from the lower layers toward the higher layers. Within each layer, we follow the topological order of underlying modules. We provide a total allowed increase of loss ratio $r$ as an input to the Algorithm 2. The hyper-parameter $r$ depends on the efficiency and efficacy trade-off which users are willing to pay. The larger the value $r$, the faster approximation we get at the cost of lower accuracy. We then distribute $r$ into each module $R_{l,i}$ (allowed loss increase ratio of $i$-th module of $l$-th layer in Algorithm 2). The distribution from $r$ to each $R_{l,i}$ is based on the observed inference time of each module $E_{l,i}$ (observed empirical inference time of $i$-th module of $l$-th layer in Algorithm 2). The longer a module takes to compute, the more budget is allocated. Overall, total allowed loss $r$ and the distributed loss ratio for each module $R_{l,i}$ fulfil the equality $(1 + r) = \prod_l \prod_i (1 + R_{l,i})$. For each module, if the approximation with certain rank used won't increase the loss over the ratio $(1 + R_{l,i})$, we will use that rank to approximate the module and move on to the next module. The pseudo code is also provided in the Appendix to illustrate the process.

---

**Algorithm 1** Data-Aware Low-rank Compression of feed-forward layer.

---

**Input:** rank $k$, training data $D_{train}$, Original weight matrix $W$, Prediction Model $M$.
**Output:** Low-rank Approximation $U^*, V^*$.
X = {}
**for** all batches $x_b$ in $D_{train}$ **do**
    Feed the batch of training data $x_b$ into $M$ and extract the representation $x$. $x$ is the representation which will be multiplied with $W$ as in (1).
    Append $x$ to X.
**end for**
Given $X, k$ and $W$, solve the optimal low-rank matrices $U^*, V^*$ by (5).

---

---

**Algorithm 2** Overall Low-rank Model Approximation Algorithm

---

**Input:** training data $D_{train}$, original weight matrix $W$. prediction Model $M$, total allowed loss increase ratio $r$, Observed inference time $E$, Search grids of ranks for each module $G$, original Training loss $L$.
**Output:** Low-rank Model $\hat{M}$.

\# Distribute allowed ratio r into each module by $E$
$E_{min} \leftarrow \arg\min_{l,i} E_{l,i}$
$E_{l,i} \leftarrow \frac{E_{l,i}}{E_{min}}$
$E_b \leftarrow exp(\frac{log(1+r)}{\sum_{l,i} E_{l,i}})$
$R_{l,i} \leftarrow E_b^{E_{l,i}} - 1$
**for** $l = 1, \cdots$ , total layers **do**
    **for each** module $m_i \in M_l$ **do**
        $W_{l,i} \leftarrow l$-th layer parameter of module $m_i$
        (e.g., 2nd feed-forward matrix in first layer.)
        **for** $i = 1, \cdots, |G_{l,i}|$ **do**
            $k \leftarrow G_{l,i}$
            $U, V \leftarrow$ Algorithm 1 $(k, D_{train}, W_{l,i}, M)$
            $\hat{M} \leftarrow M$ with $W_{l,i}$ replaced by $U, V$.
            Evaluate new loss $L_{new} = \hat{M}(D_{train})$
            **if** $L_{new}/L < 1 + R_{l,i}$ **then**
                $M \leftarrow \hat{M}$
                break;
            **end if**
        **end for**
    **end for**
**end for**

---

# 4  Experimental Results

## 4.1  Experimental Setup

We evaluate DRONE on both LSTM and transformer-based BERT models. For LSTMs, we train a 2-layer LSTM-based language model from scratch with hidden sizes 1500 on Penn Treebank Bank (PTB) dataset. For BERT models, we evaluate the pre-trained BERT models on GLUE tasks. Various pre-trained models are offered in the open source platform [40]. For BERT models, we use BERT-base models and it contains 12 layers of the same model structure without sharing parameters. Each layer contains an attention module with hidden size 768 and 12 channels, a small $768 \times 768$ Feed-forward (FF) layer followed by 2 larger FF layers ($768 \times 3072$ and $3072 \times 768$). As shown in Figure 1, these four components consume the most computational time in the BERT-base models.

For the baseline methods, most of the existing work pertaining to low-rank approximation [21, 32] leverages SVD in part of the compression procedure. Therefore, our baseline comparison will be the SVD approximation, and our work aims to provide an improvement over SVD. We also include the state-of-the-art distillation methods TinyBERT [15] in the comparison and show that the proposed method can be combined with it to further improve the performance. TinyBERT reduces the model

Table 1: The experimental results of running pret-rained BERT-base model on natural language inference tasks (Glue dataset). Each task has its own metric for performance measurement. Accuracy (SST-2, QNLI, RTE and WNLI), F1/Accuracy (MRPC and QQP), Matthew's correlation (CoLA), Matched accuracy/Mismatched accuracy (MNLI) and Person/Spearman correlation (STS-B) are used respectively. All the DRONE results are within 3% accuracy loss and show that DRONE can accelerate the whole BERT model across different tasks and devices.

| Methods | MNLI | QQP | SST-2 | QNLI | MRPC | RTE | CoLA | STS-B |
|---|---|---|---|---|---|---|---|---|
| Original | 84.3 | 90.9 | 92.3 | 91.4 | 89.5 | 72.6 | 53.4 | 87.8 |
| SVD | 74.4 | 50.8 | 73.1 | 52.2 | 63.8 | 47.3 | 12.4 | 33.6 |
| DRONE | 82.0 | 89.4 | 90.0 | 88.5 | 86.7 | 70.0 | 52.5 | 85.8 |
| DRONE-Retrain | 82.6 | 90.1 | 90.8 | 89.3 | 88.0 | 71.5 | 53.2 | 87.8 |
| CPU Speedup Ratio | 1.60x | 1.25x | 1.64x | 1.20x | 1.92x | 1.31x | 1.33x | 1.52x |
| GPU Speedup Ratio | 1.28x | 1.38x | 1.45x | 1.28x | 1.56x | 1.33x | 1.29x | 1.57x |

into 4 layers of attention dimension 312 with 12 channels, and the FF layers are downsized to $312 \times 1200$. As we mentioned above, all the approximation methods need to consider efficiency and efficacy trade-off. In this paper, we follow previous literature [15, 5] and report the approximation results with about 3% loss in accuracy to compare the performance of all methods.

In real-world applications, NLP models are mostly evaluated on mobile devices or servers with multiple hardware accelerators. Thus, we measure the inference speed on both CPU (Intel(R) Xeon(R) CPU E5-2640 v4 @ 2.40GHz) and GPU (GeForce GTX 1080 Ti) devices. All the experiments are repeated 10 times. The average single sequence prediction inference time in milliseconds is reported in the results. We want to emphasize that unlike many of the literature [38, 17, 7], which reported speedup only in the attention layers, our results reflect end-to-end speedup including both attention module and feed-forward layers. To perform the approximation, empirically we found randomly sub-sample 10% of the training data suffices to provide good results. Using more data can only provide limited performance boost but comes at a higher cost of longer preprocessing time. Thus, we will use 10% random sample of the training data to perform the experiments. After the proposed data-aware low-rank distribution, we slightly fine-tune the model to further improve the performance. We use a relatively smaller learning rate $10^{-7}$ and retrain 1 epoch on the sub-sampled training data to complete the fine-tuning procedure.

## 4.2 Results of BERT Models on GLUE Dataset

We summarize the results of DRONE on GLUE tasks in Table 1. Detailed inference time of each component of the compressed Transformer model is listed in the Appendix. Detailed inference time of an uncompressed BERT-BASE model can be found in Figure 1. We observe that each task exhibits different difficulty. The best acceleration we can achieve is nearly twice as fast (1.92x) with less than 2% accuracy loss after retraining (on the MRPC). In addition, DRONE achieves 1.52x acceleration without accuracy loss on the STS-B task. By applying the same selected rank for each module with SVD method, we can observe that the performance drops significantly. This shows that the matrices within the model is generally not low-rank; thus the direct low-rank approximation without considering data distribution does not work. On GPU, we see that the acceleration is more or less the same as on CPU except MNLI and MRPC tasks. This is due to the fact that GPU uses massive parallelism and low-rank approximation introduces a sequential computation which might hinder the speedup depending on the size of the matrices and ranks used. To resolve this problem, low-level cuda code optimization is needed and system researchers have studied the problem [29], which is out of the scope of the present work. Despite this, we can still observe that DRONE performs better on QQP, RTE and STS-B and it provides about 1.5x acceleration for various tasks on GPU. An example of ranks used in SST-2 is listed in Appendix F.

## 4.3 Combination with Model Size Reduction Methods

Our proposed DRONE is a general low-rank approximation technique, and it is complementary to many other model compression methods. To illustrative its power, we now demonstrate that DRONE can be combined together distillation. The discussion of combination with Quantization is left in the Appendix. Distillation methods compress the underlying model into a smaller one without losing much accuracy. Distilled models are much smaller in number of layers or hidden dimension, resulting in a smaller model size and faster inference time. As shown in the Table 2, TinyBERT, one of the most competitive distillation methods, indeed achieves good performance within 3% accuracy loss for some of the GLUE tasks. Due to the fact that the computation inside the distilled model is still

full matrix computation, DRONE can be applied to find data-aware low-rank approximation of these smaller matrices. Results are summarized in Table 2. As we can see combining DRONE with the distillation method further reduces the inference time without sacrificing accuracy. In particular, on the SST-2 task DRONE + TinyBERT speeds the inference time from 11.7x to 15.3x on CPU while achieving the same accuracy as the TinyBERT. Similarly, DRONE + TinyBERT speedups GPU results with 10.9x STS-B and 9.7x on SST-2 with competitive performance. These results again show that the proposed method has the potential to be applied under various scenarios and hardware devices to achieve a better model inference time speedup.

Table 2: The average inference time (in milliseconds) in comparison to distilled models on CPU and GPU. The unit is in millisecond. The results show that DRONE can be combined with distillation to further improve the performance. Compared to the state-of-the-art distillation method, the speedup ratio increases from 11.4x to 14.2x on STS-B and from 11.7x to 15.3x on SST-2.

| Tasks | Models | CPU-speedup | GPU-speedup | Accuracy (%) |
|---|---|---|---|---|
| STS-B | BERT | 1x | 1x | 87.8 |
| | TinyBERT | 11.4x | 8.6x | 86.9 |
| | DRONE +TinyBERT | **14.2x** | **10.9x** | **87.0** |
| RTE | BERT | 1x | 1x | 72.6 |
| | TinyBERT | 1.8x | 1.9x | 70.8 |
| | DRONE +TinyBERT | **2.1x** | **2.2x** | **71.7** |
| MRPC | BERT | 1x | 1x | 89.5 |
| | TinyBERT | 11.6x | 7.8x | 86.3 |
| | DRONE +TinyBERT | **12.3x** | **8.6x** | **86.7** |
| SST-2 | BERT | 1x | 1x | 92.3 |
| | TinyBERT | 11.7x | 8.4x | **90.7** |
| | DRONE +TinyBERT | **15.3x** | **9.7x** | **90.7** |

Table 3: Illustration of SVD fine-tuning on MRPC, RTE, CoLA and STS-B. Using the same rank as the proposed DRONE method, SVD accuracy will drop significantly after the approximation. After fine-tuning done on the SVD approximation, the accuracy could be recovered for some tasks (e.g., MRPC), but SVD + Retrain still perform much worse than DRONE across all the tasks.

| Models | MRPC | RTE | CoLA | STS-B |
|---|---|---|---|---|
| BERT | 89.5 | 72.6 | 53.4 | 87.8 |
| DRONE-Retrain | **88.0** | **71.5** | **53.2** | **87.8** |
| SVD | 63.8 | 47.3 | 12.4 | 33.6 |
| SVD-Retrain | 85.8 | 63.5 | 24.4 | 66.3 |

## 4.4 Comparison to Structured Pruning Methods.

Instead of compressing model after training to accelerate inference time, another line of research called Structured Pruning tried to learn the low-rank structure during training of the model to save both training and inference time simultaneously. This raises the question if post-processing such as DRONE is necessary if no further compression is required once we can get a small model after training. Thus, it's worth comparing DRONE with the state-of-the-art Structure Pruning method [39]. In [39], attention modules are not approximated by low-rank matrices. To make the comparison fair, we apply DRONE on base models except the attention module and keep others the same. For MRPC, DRONE achieves 1.58x speedup with performance drop from 89.5 to 89.4. [39] achieves 1.43x with performance 88.61. For SST-2, DRONE achieves 1.41x speedup without sacrificing performance (92.3). [39] also achieves about 1.41x speedup with the performance 92.09. Thus, we can see that DRONE performs better than structured pruning.

One further question is that if we can use the idea of DRONE to do fast training? We conducted DRONE on the pre-trained RTE task before fine-tuning to get a low-rank structure, and then apply the regular end-to-end training over this compressed model. We found out this procedure with directly using the same rank as in our experiments ( with 1.38x speedup) can only achieve 68.2 accuracy. But if we reduce the compression ratio into 1.2x training time speedup, this procedure can give us 72.9 accuracy. On the other hand, the same procedure with SVD as the initialization of low-rank structure can only get 52.1 accuracy. This preliminary experiment shows that in addition to inference time acceleration, DRONE also has the potential to be applied in the training, but directly transport

DRONE into training can not lead to the optimal result. How to improve low-rank training is an interesting future direction.

## 4.5 Additional Experiments on Large-Scale Models and Language Generation Tasks

Concerns might be raised that if DRONE can also be generalized to other scenarios such as larger models or other NLP tasks. To validate DRONE on larger models. We conduct experiments on RTE dataset with BERT-LARGE model. BERT-LARGE doubled number of layers and the dimension is increased from 768 to 1024. Average inference time for a data increased to 1405ms and it achieves accuracy 74. The overall result is 72.9 with inference time 1018ms (1.38x speedup). This result is comparable to our BERT-BASE result (1.31x speedup). To validate DRONE on other NLP tasks, We conducted the method on the machine translation task via OpenNMT. It provides a 2-layer transformer model on en-de translation. On the transformer part, DRONE achieves 1.76x speedup with BLEU from 33.47 to 33.26. Through these two additional experiments, we can validate that the proposed DRONE is generic in the sense that so long as the underlying model is composed of matrix computation, DRONE can compress the model regardless of model sizes and target tasks.

## 4.6 Can we directly learn low-rank structures by end-to-end training?

From an optimization perspective, a natural question to ask is whether the same optimal low-rank structure could be learned by end-to-end fine-tuning once the rank is decided. We conduct experiments on 4 tasks to verify this, and the results are summarized in Table 3. We start by performing DRONE on the task to achieve the desired accuracy, and perform SVD with the same set of ranks. Accuracy of SVD drops significantly for all tasks. We then fine-tune hyper-parameters as in [40][1] to fine-tune the above SVD results. After fine-tuning, the accuracy improves across all tasks, but none of it can reach the same performance as DRONE. This shows that due to the difficulty of optimizing a non-convex objective function, fine-tuning the SVD result may not achieve the best low-rank result. On the other hand, the proposed DRONE method under the the optimization problem (3) can obtain the provably optimal low-rank approximation at a much lower computational cost than the fine-tuned SVD.

## 4.7 Pre-processing of DRONE is not too costly

DRONE accelerates inference speed at the cost of a pre-processing step. Thus, It's natural to ask if DRONE will take long pre-processing time. Given the rank, prep-rocessing of DRONE has a one-time distribution extraction plus low-rank solving of equation (3). Depending on training data size, first stage takes 2 mins (RTE) to 20 mins (MNLI). Second stage has 2 SVD computations and is about 5-10 mins. The matrix size involved is limited as we subsample training data. SVD costs about 3 mins and retrain costs from 5 mins to 2 hours depending on training data size. Thus, pre-processing of DRONE is about the same order as SVD but with much better performance. Distillation such as TinyBERT firstly has a general distillation of BERT-base on large data used to train original BERT, followed by task-specific distillation. Despite task-specific one is rather fast (15 mins to 2hrs), first stage takes a few days for a single GPU. Overall, DRONE is not costly compared to other methods.

## 5 Conclusions

In this paper, we propose DRONE, a data-aware low-rank approximation, to achieve a better low-rank approximation in BERT models. DRONE leverages the fact that data distribution in NLP tasks usually lies in a lower-dimensional subspace. By considering the data distribution, we propose a data-aware low-rank approximation problem and provide a closed-form solution. Empirical results validate that DRONE can significantly outperform the vanilla-SVD method, and can achieve at least $20\%$ acceleration with less than $3\%$ accuracy loss. When DRONE is combined with distillation methods, it further achieves up to $15.3$ times acceleration with less than $2\%$ accuracy loss.

---

[1]https://huggingface.co/transformers/v2.1.1/examples.html#glue

# 6 Acknowledgement

We would like to thank the anonymous reviewers for their helpful comments. This work is supported in part by NSF under IIS-1901527, IIS-2008173, IIS-2048280.

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
