# Appendix for:
## Data-Aware Low-Rank Compression for Large NLP Models

## A  Proof of Theorem 1

**Theorem 1.** *Assume rank($W$) $= r$ and rank($X$) $= t$. The closed form solution $M^*$ of the optimization problem in equation 3 is*

$$M^* = V_{W,r} S_{W,r}^{-1} Z_k S_{X,t}^{-1} U_{X,t}^T, \tag{9}$$

*where $Z_k$ is the rank-$k$ truncated SVD of $Z = S_{W,r} V_{W,r}^T U_{X,t} S_{X,t}$.*

*Proof.* We firstly consider the unconstrained problem:

$$
\begin{aligned}
M^* &= \arg\min_M \|WX - WMX\|_F^2 \\
&= \arg\min_M \|U_W^T WXV_X - U_W^T WMXV_X\|_F^2 \\
&= \arg\min_M \|S_W V_W^T U_X S_X - S_W V_W^T MU_X S_X\|_F^2,
\end{aligned}
$$

where the second equality holds due to the fact that $U_W$ and $V_X$ are orthonormal matrices. Note that we could expand the term $S_W V_W^T U_X S_X$ as:

$$
\begin{aligned}
S_W V_W^T U_X S_X &= \begin{bmatrix} S_{W,r} & 0 \\ 0 & 0 \end{bmatrix} \begin{bmatrix} V_{W,r}^T \\ \bar{V}_{W,r}^T \end{bmatrix} \begin{bmatrix} U_{X,t} & \bar{U}_{X,t} \end{bmatrix} \begin{bmatrix} S_{X,t} & 0 \\ 0 & 0 \end{bmatrix} \\
&= \begin{bmatrix} S_{W,r} V_{W,r}^T & 0 \\ 0 & 0 \end{bmatrix} \begin{bmatrix} U_{X,t} S_{X,t} & 0 \\ 0 & 0 \end{bmatrix} \\
&= \begin{bmatrix} S_{W,r} V_{W,r}^T U_{X,t} S_{X,t} & 0 \\ 0 & 0 \end{bmatrix}.
\end{aligned}
$$

Similarly, we will have

$$
S_W V_W^T MU_X S_X = \begin{bmatrix} S_{W,r} V_{W,r}^T MU_{X,t} S_{X,t} & 0 \\ 0 & 0 \end{bmatrix}.
$$

Therefore, we continue above unconstrained problem as:

$$
\begin{aligned}
M^* &= \arg\min_M \|S_W V_W^T U_X S_X - S_W V_W^T MU_X S_X\|_F^2 \\
&= \arg\min_M \left\| \begin{bmatrix} S_{W,r} V_{W,r}^T U_{X,t} S_{X,t} - S_{W,r} V_{W,r}^T MU_{X,t} S_{X,t} & 0 \\ 0 & 0 \end{bmatrix} \right\|_F^2 \\
&= \arg\min_M \|S_{W,r} V_{W,r}^T U_{X,t} S_{X,t} - S_{W,r} V_{W,r}^T MU_{X,t} S_{X,t}\|_F^2. \\
&= \arg\min_M \|Z - S_{W,r} V_{W,r}^T MU_{X,t} S_{X,t}\|_F^2.
\end{aligned}
$$

The above minimization problem obtains the optimal value if $S_{W,r} V_{W,r}^T MU_{X,t} S_{X,t}$ equals the rank-$k$ truncated SVD of Z by the fundamental property of SVD decomposition. Thus, we will have:

$$
\begin{aligned}
Z_k &= S_{W,r} V_{W,r}^T M^* U_{X,t} S_{X,t} \\
\implies M^* &= V_{W,r} S_{W,r}^{-1} Z_k S_{X,t}^{-1} U_{X,t}^T.
\end{aligned}
$$

$\square$

# B   An algorithm to Search of Ranks under DRONE

The input to Algorithm 2 consists of training data, the model with all parameters of weight matrices and original training loss. In addition, a pre-defined search grid is also necessary. Taking $W \in R^{768 \times 768}$ as an example, we can perform a grid search for a proper low rank $k$ over $[1, 768]$ such as $\{96, 192, 288, 384, \ldots, 768\}$. The finer the grid, the more compressed model we could get at the cost of longer running time of the DRONE method. With these input parameters, we firstly distribute the total allowed loss into each individual module. We then iteratively apply Algorithm 1 following the computational sequence illustrated in Figure 1. For each module, we search the rank $k$ by going through the grid. If the approximated result will not increase the allowed loss increase ratio of the component, we will end the search and tie the found rank to the component and move on. The procedure will continue until all components are compressed. The whole process could guarantee us that the final loss $L'$ of the compressed model $\hat{M}$ would not be greater than $(1 + r)L$, where $L$ is the original loss before approximation.

---

**Algorithm 2** Overall Low-rank Model Approximation Algorithm

---

**Input:** training data $D_{train}$, original weight matrix $W$. prediction Model $M$, total allowed loss increase ratio $r$, Observed inference time $E$, Search grids of ranks for each module $G$, original Training loss $L$.
**Output:** Low-rank Model $\hat{M}$.

\# Distribute allowed ratio r into each module by $E$
$E_{min} \leftarrow \arg\min_{l,i} E_{l,i}$
$E_{l,i} \leftarrow \frac{E_{l,i}}{E_{min}}$
$E_b \leftarrow exp(\frac{log(1+r)}{\sum_{l,i} E_{l,i}})$
$R_{l,i} \leftarrow E_b^{E_{l,i}} - 1$
**for** $l = 1, \cdots,$ total layers **do**
    **for each** module $m_i \in M_l$ **do**
        $W_{l,i} \leftarrow l$-th layer parameter of module $m_i$
        (e.g., 2nd feed-forward matrix in first layer.)
        **for** $i = 1, \cdots, |G_{l,i}|$ **do**
            $k \leftarrow G_{l,i}$
            $U, V \leftarrow$ Algorithm 1 $(k, D_{train}, W_{l,i}, M)$
            $\hat{M} \leftarrow M$ with $W_{l,i}$ replaced by $U, V$.
            Evaluate new loss $L_{new} = \hat{M}(D_{train})$
            **if** $L_{new}/L < 1 + R_{l,i}$ **then**
                $M \leftarrow \hat{M}$
                break;
            **end if**
        **end for**
    **end for**
**end for**

---

# C   Efficiency and Efficacy Trade-off Graph

In this paper, we mentioned that we report the result of 3% accuracy drop as the performance of the baseline methods and DRONE. However, as we mentioned above that all the approximation methods need to consider efficiency and efficacy trade-off. 3% is chosen according to the literature. Here, we show two exemplar graph on MRPC and SST-2 task to demonstrate two facts. First, it's indeed a trade-off between the efficiency and efficacy as the speedup ratio goes higher at the cost of lower accuracy. Second, we want to point out that this trade-off relationship is not linear, and different task might have different characteristics. Thus, in the real application, users need to decide what's the best cutoff to use. We also want to point out that this 3% accuracy drop comparison is fair to all baseline methods. We could have chose another cutoff like 1% accuracy with lower speedup ratio to report, but this won't help too much when comparing different baseline methods.

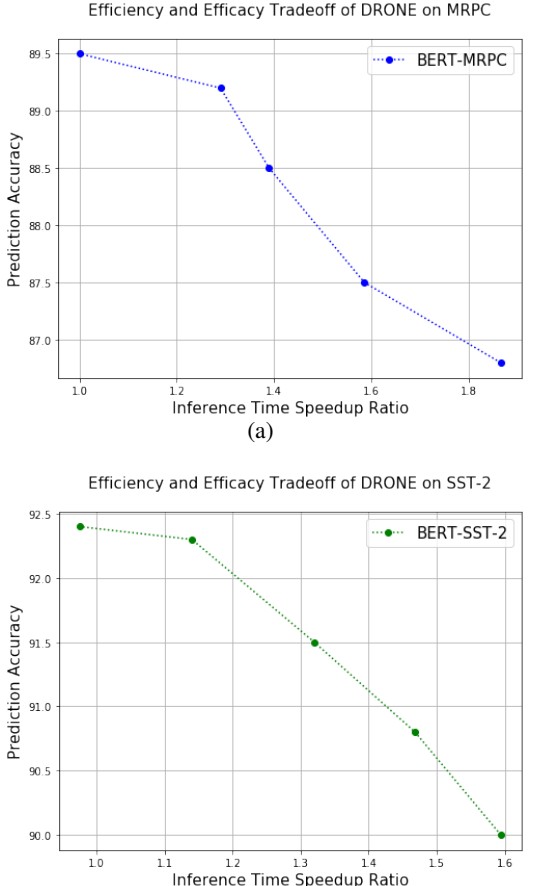

Figure 3: Illustration of efficiency and efficacy trade-off. Each point in this graph represents a specific ratio of training loss increase after approximation.

# D Detailed results

## D.1 LSTM result

A 2-layer LSTM model is composed of two large matrices layers and one large softmax layer. Additional processing time includes applying activation functions, softmax function and computing the updated hidden representation. The detailed inference time in each layer is summarized in Table 4. We could observe that the overhead of the computation will be greatly incurred on GPU. Thus, despite the matrix is much smaller and well approximated by DRONE, the overall acceleration on GPU is less.

## D.2 Transformer result

For BERT models, we use BERT-base models and it contains 12 layers of the same model structure without sharing parameters. Each layer contains an attention module with hidden size 768 and 12 channels, a small $768 \times 768$ Feed-forward (FF) layer followed by 2 larger FF layers ($768 \times 3072$ and $3072 \times 768$). As shown in Figure 1, these four components consume the most computational time in the BERT-base models. The detailed average inference time of each module is summarized in Table 5 for CPU and Table 6 for GPU. There are two important points to note.

Firstly, we could see that attention module is not the bottleneck at all under the normal size of context (128). Therefore, many works on accelerating attention module alone would not improve the overall inference time of the module except a very long sequence appears. The necessity of the long sequence is out of the domain of this paper and what we want to show is that the proposed DRONE would work on both attention and feed-forward layer, which collectively could accelerate the real(overall) inference time.

Secondly, we could observe that the FF2 layer could be accelerated most. A plausible reason could be that the input dimension to the FF2 layer is in a larger dimension (3072) than all the other layers (64 or 768). When the input distribution actually lies in a lower-dimensional space, there is much more room for FF2 layer to be compressed and accelerated by the data-aware low-rank method.

Table 4: The average inference time of each component in the model of 2-layer LSTM model. Both proposed methods and SVD use same ranks so the inference time is approximately the same. The unit is in millisecond and the number in parenthesis shows the ratio respective to the overall inference time.

| Device | Models | LSTM-1 | LSTM-2 | Softmax | Others | Total Time | Perplexity |
|---|---|---|---|---|---|---|---|
| | PTB-Large | 1.27ms | 1.30ms | 1.09ms | 0.13ms | 3.79ms | 78.32 |
| | PTB-Large-SVD | - | - | - | - | - | 81.09 |
| CPU | PTB-Large-SVD-Retrain | - | - | - | - | - | 80.89 |
| | PTB-Large-DRONE | - | - | - | - | - | 80.87 |
| | PTB-Large-DRONE-Retrain | 0.24ms | 0.34ms | 0.42ms | 0.11ms | 1.11ms(3.4x) | 79.01 |
| | PTB-Large | 0.019ms | 0.018ms | 0.015ms | 0.32ms | 0.11ms | 78.32 |
| | PTB-Large-SVD | - | - | - | - | - | 81.09 |
| GPU | PTB-Large-SVD-Retrain | - | - | - | - | - | 80.89 |
| | PTB-Large-DRONE | - | - | - | - | - | 80.87 |
| | PTB-Large-DRONE-Retrain | 0.01ms | 0.01ms | 0.015ms | 0.055ms | 0.09ms(1.2x) | 79.01 |

Table 5: The detailed average inference time (in milliseconds) on CPU of each component in the model by retrained DRONE.

| Tasks | Self-Attention | Feed-Forward 0 | Feed-Forward 1 | Feed-Forward 2 | Others | Total Time |
|---|---|---|---|---|---|---|
| MNLI | 122.7 | 19.5 | 78.5 | 46.1 | 4.2 | 271.0 |
| QQP | 131.5 | 29.9 | 99.2 | 66.5 | 5.8 | 333.0 |
| SST-2 | 100.5 | 24.7 | 79.3 | 54.5 | 4.5 | 263.5 |
| QNLI | 128.3 | 28.4 | 111.0 | 79.0 | 5.9 | 352.6 |
| MRPC | 82.6 | 12.8 | 89.4 | 38.2 | 2.4 | 225.4 |
| RTE | 116.0 | 25.6 | 85.4 | 62.3 | 3.4 | 292.7 |
| CoLA | 108.2 | 22.7 | 93.1 | 70.8 | 3.4 | 298.2 |
| STS-B | 109.1 | 19.3 | 90.8 | 53.0 | 4.0 | 276.2 |

Table 6: The detailed average inference time (in milliseconds) on GPU of each component in the model by retrained DRONE.

| Tasks | Self-Attention | Feed-Forward 0 | Feed-Forward 1 | Feed-Forward 2 | Others | Total Time |
|-------|----------------|----------------|----------------|----------------|--------|------------|
| MNLI | 0.94 | 0.26 | 0.76 | 0.60 | 0.003 | 2.56 |
| QQP | 0.92 | 0.24 | 0.64 | 0.52 | 0.001 | 2.32 |
| SST-2 | 0.85 | 0.25 | 0.59 | 0.52 | 0.005 | 2.22 |
| QNLI | 0.91 | 0.25 | 0.72 | 0.60 | 0.001 | 2.48 |
| MRPC | 0.89 | 0.22 | 0.57 | 0.43 | 0.001 | 2.11 |
| RTE | 1.02 | 0.29 | 0.60 | 0.59 | 0.008 | 2.51 |
| CoLA | 0.93 | 0.25 | 0.68 | 0.61 | 0.002 | 2.47 |
| STS-B | 0.83 | 0.2 | 0.57 | 0.42 | 0.002 | 2.02 |

# E    Combination with Quantization Methods

Distillation in practice achieves the STOA without extra hardware accelerator, so it serves as a good target to show how DRONE can be combined with other methods. The benefit of quantization/pruning can only be shown when a ASIC/FPGA accelerator is provided. Since we don't have one, we can't only use the software to simulate. We can apply any Quantization scheme and empirically show combined method can achieve a competitive accuracy with lower bit bandwidth. Algorithmically, we combine DRONE with vanilla quantization with fixed precision which gets 87.5 (vs 89.5 on MRPC) and 51.0 (vs 53.4 on CoLA) with 12 bits(vs 32 bits). Thus we can hypothesize that with the hardware accelerator, there could be at least further 3x speedup when DRONE is combined with Quantization methods.

# F    An example of ranks used in SST-2

Below are the ranks obtained by performing DRONE in SST-2 dataset. The order is from the bottom layer to the top layer. Full rank of all the matrices are 768.

Attention layer: [192, 384, 192, 768, 768, 192, 768, 768, 192, 192, 768, 192],

Feed-Forward 0 : [288, 768, 96, 192, 288, 192, 768, 768, 96, 96, 288, 96],

Feed-Forward 1: [96, 96, 768, 768, 288, 288, 768, 768, 96, 96, 288, 96],

Feed-Forward 2 : [192, 192, 768, 288, 768, 768, 768, 192, 96, 192, 96, 96].

## G   Python Pseudo Code for Solving equation (3)

Listing 1: The python function to solve the equation (3).

```python
import numpy as np

def OPTsolver(x,y,k):
    '''
    compute  the  best  rank  k  projection  M  such  that  \| x*y' - x*M*y '\|_{F}
    is  minimized
    x \in  shape  n  x  d
    y \in  shape  m  x  d

    '''
    xSS = np.matmul(x.transpose(),x)
    kSS = np.matmul(y.transpose(),y)
    U1,S1,V1 = np.linalg.svd(xSS,False)
    S1 = S1 ** 0.5
    I1 = np.eye(S1.shape[0])
    U2,S2,V2 = np.linalg.svd(kSS,False)
    S2 = S2 **0.5
    I2 = np.eye(S2.shape[0])
    YK = np.dot(np.dot(I1*S1,V1),np.dot(V2.transpose(),I2*S2))
    U,S,V = np.linalg.svd(YK,False)
    L = np.dot(V1.transpose(),I1*(1/S1))
    R = np.dot(I2*(1/S2),V2 )
    M = np.dot(U[:,:k]*S[:k],V[:k,:])
    return L,R,U,S,V
```

## H   Python Pseudo Code of Rank Searching

Listing 2: A mixed of real code and pseudo code to illustrate the search algorithm.

```python
import os
import numpy as np
import torch
import subprocess as sp

cuda_num = 7
n_heads = 12
total_layer = 12

prev_loss = .11159391902588509 # Initial Loss
the_model_name = 'bertSST2'

time_attn = 117.5 # Empirical Inference Time on Attention Module
time_0 = 34.27 # Empirical Inference Time on Attention FFL Module
time_1 = 133.11 # Empirical Inference Time on Feedforward 1 layer
time_2 = 128.84 # Empirical Inference Time on Feedforward 2 layer
minimal_time = min(time_attn,time_0,time_1,time_2)
multiplier = (time_attn+time_0+time_1+time_2)/(minimal_time)
tolerant = 2. # allowed loss increase ratio. $r$ in Algorithm 2.

# Code to Distribute the $r$ into individual Modules.
# The distribution depends on empirical inference time of each module and number of layers.
basic_tolerance = np.exp(np.log(tolerant)/multiplier)
tol_attn = np.exp(np.log(basic_tolerance**(time_attn/minimal_time))/n_layer)
tol_0 = np.exp(np.log(basic_tolerance**(time_0/minimal_time))/n_layer)
tol_1 = np.exp(np.log(basic_tolerance**(time_1/minimal_time))/n_layer)
tol_2 = np.exp(np.log(basic_tolerance**(time_2/minimal_time))/n_layer)
```

```
#### Omitted Code ###
# This part of the code is to change some parameters of the underlying hugginface framework
in order to extract the training distribution X of each module from the model.
### Omitted Code ###

for i in range(total_layer):
    for each module in the layer: # This line is pseudo code for clarity reason.
        # This part of the code extracts $R_{l,i}$(named the_tol here) in Algorithm 2.
        if save_symbol == "E":
            the_tol = tol_attn
        elif save_symbol == "F0":
            the_tol = tol_0
        elif save_symbol == "F1":
            the_tol = tol_1
        else:
            the_tol = tol_2
        # Update the allowed increase of loss
         prev_loss = prev_loss * the_tol

        # initial search rank for Attention(16) and FFL layers(96)

        rank = 16 if save_symbol == "E" else 96

        # Maximal rank
        tps = 64 if save_symbol == "E" else 768 specified in the original models.
        while rank <= tps:
            ### Omitted Code ###
            ## Write the tried rank into hugginface framework##
            ### Omitted Code ###

            # This line run the inference in the command line
            os.system(python run_glue.py --model_type bert --task_name SST-2)

            with open('/tmp/tmp0','r') as file:
                data = file.readlines()
            new_r = float(data[-1])
            if  new_r < prev_loss:
                break
            if save_symbol == "E": # Attention module, we increase search rank 16 at a time.
                rank += 16
            else:
                #rank += 96 # For FFL layer, we increase search rank 96 a time.
                if rank ==   384:
                    rank = 768
                    break
                else:
                    rank += 96

    ### Omitted Code ###
    # This part of code update the model #
```