# OpenReview forum: "DRONE: Data-aware Low-rank Compression for Large NLP Models"
_NeurIPS.cc/2021/Conference — NeurIPS 2021 Poster_

### Official Review · Reviewer_o4hz · 2021-07-13

**Rating:** 6
**Confidence:** 3

**Summary:**

This paper proposes DRONE, a data-aware low-rank model compression method designed specifically for large language models, e.g., BERT. The method is motivated by an observation that the weight matrices in a pre-trained BERT model are generally not low rank, but the data/intermediate results usually lie in a low-rank space. DRONE leverages the above observation and arrives at a close form solution to approximate the layer weights using low-rank factorization. Theoretical analysis is provided and experimental results are shown to demonstrate that DRONE successfully accelerate the inference speed of BERT model on a various range of down stream tasks.

**Ethical Concerns:**

I do not believe that this paper raises any ethical concerns.

**Limitations And Societal Impact:**

My major suggestions are summarized in the “Cons” of “Main review”. Please take a close look on it. Moreover, as the state-of-the-art architectures in computer vision and NLP fields tend to be unified, e.g., Transformers/pure MLP networks. It would be important to include both NLP and computer vision results, e.g., ViT/MLP-Mixer on CIFAR and ImageNet to show that the method generalize well.

I am happy to increase my overall evaluation scores if my concerns are addressed.

**Main Review:**

Pros:
1. The paper is well-written, and the research direction of compressing the large language models and improving the inference speed is promising.
2. The experimental results are promising. I am convinced that DRONE can work well in practice.

Cons:
My major concern on the proposed method is about its novelty. The data-aware low-rank compression method has been explicitly proposed in PCN [1] (which is not even cited by this paper). In my view, DRONE is almost identical to PCN apart from the fact that PCN has not been deployed in language models yet. The authors are expected to elaborate the novelty of DRONE further. Other concerns are summarized as follows:
1. Decomposing one layer into two small low-rank layers can effectively increase the depth of the model (which reducing width), which is known to make the model harder to train. In such a sense, how dose the training behavior of DRONE look like? And how does that compare to the vanilla BERT training.
2. On the memory perspective of view, deploying low rank approach reduces the model size. But the activation values for each factorize layers need to be saved. How does the memory footprint look like?
3. It has been studied that starting from a “warm-up” stage of full-rank model can be helpful for the low-rank model [2-3]. Does it also help DRONE?
4. In the current draft, DRONE is not compared with any other pruning/sparsification based methods. How does DRONE compare to the structured pruning method, e.g., [3]?
5. The current experiments only study BERT models, whose scales are not that large. How does DRONE perform on GPT-2/ViT/MLP-Mixer models?
6. It is easy to imagine that one can use DRONE to speed up the BERT pre-training stage. Adding even an preliminary result on such a direction can be really helpful.
Minor issues:
Missing references: [1-2], [4].

[1] https://arxiv.org/abs/2006.13347
[2] https://arxiv.org/abs/2103.03936
[3] https://arxiv.org/pdf/1910.04732.pdf
[4] https://openreview.net/forum?id=KTlJT1nof6d

**Time Spent Reviewing:**

2.5

---

> ### Author Response · Authors · 2021-08-10
> **Thank reviewer for valuable opinions and we will reply to each individual point**
>
> - On difference between PCN and novelty
>
> We thank the reviewer for pointing out PCN! We agree that it’s also a data-aware  compression method and we will revise our writeup, but below are important points.
>
> 1.For input transformation it uses PCA. So the obtained transformation does not consider the current weight matrix W. Thus, it’s a special case of DRONE in eq (3) with W to be an Identity matrix. An important novelty of DRONE is that we not only consider distribution X, but also take the W into consideration.
>
> 2.Adding output transformation into consideration, despite the dimension being reduced, there is no rank constraint on the final W. Thus, W is not a low-rank structure and DRONE will return a low-rank W to further accelerate the computation. In this view, PCN is orthogonal to DRONE and this shows another novelty of DRONE that it can be combined with other acceleration methods.
>
> 3.PCN uses PCA as a reasonable motivation to compress the model; however, it’s not guaranteed that the obtained transformation is the optimal; whereas, DRONE formulates an approximation optimization problem and we provide the optimal solution.
> We will add these discussions to the paper.
>
>
> - On comparison to structured pruning
>
> We thank the reviewer for pointing this out and we strongly agree that structured pruning should be included. We have referred to the paper [1] and an open implementation [2]. However, our run of [2] gives us results worse than the numbers in the paper (same compression rate), so we will use numbers from paper to compare the performance and our trained model to measure inference time.
> One important difference between DRONE and [1] is that [1] focuses on model compression but not acceleration, but low-rank matrices cannot directly be transformed into faster attention computation. Specifically, key and value are mapped into different ranks so it cannot be multiplied directly. In addition, Attention is done in a multi-head fashion and there is no mapping between learned rank-1 matrices to the heads so the method in [1] cannot be used to accelerate attention computation. On the other hand, DRONE treats Q,K matrices as a unified low-rank structure so it can be used to accelerate Attention.
>
> To make the comparison fair, we apply DRONE on base models except the attention module and keep others the same. For MRPC, DRONE achieves 1.58x speedup with performance drop from 89.5 to 89.4. [1] achieves 1.43x with performance 88.61. For SST-2, DRONE achieves 1.41x speedup without sacrificing performance (92.3). [1] also achieves about 1.41x speedup with the performance 92.09. Thus, we can see that DRONE performs better than structured pruning. The benefit of structured pruning is that it can work on model pre-training but our focus is on the inference time speedup.
>
>
> - On Large Models
>
> In the early stage of the project, we have tried DRONE on BERT-Large which achieves 1.5x speedup with < 1% accuracy loss. However, we observed that most of the existing compression works in language community are comparing within BERT-base so later on we focus on the experimental results with BERT-BASE. Exploring CV models such as MLP-Mixer is an interesting direction.
>
> - On training behaviour of DRONE
>
> We focus on approximating an unfactorized model after the training stage so the underlying model has already reached a local minima. DRONE provides a low-rank framework considering both data distribution and the weight matrix in the unfactorized model. The approximation is done layer-by layer and each time the loss is limited. So it can be thought as a slight move from the minima but it’s still in a relatively smooth region and it’s easier to fine-tune after DRONE’s approximation.
>
>
>
> - On applying on pretraining stage
>
> Our focus is on approximating trained models into low-rank structure to accelerate inference time. In general, it outperformed factorization + pretraining together as we show in the comparison to structured pruning. Exploring using DRONE in pretraining is an interesting direction but it might not be that straightforward. Specifically, DRONE relies on both weight W and data X to aggregate in a subspace. However, randomly initialized model in pretraining stage doesn’t behave in this way. So what’s the timing to apply DRONE is a challenging task.
>
>
> - On warm-up stage helpfulness
>
>  Again we focus on inference time training so warm-up affects little when our base model is already a well-trained one. It’s interesting
>   to study how to incorporate DRONE into training from scratch.
>
> - On memory footprint
> Take MRPC for example, the checkpoint size is 248MB and the peak memory usage is 718MB. The corresponding full model checkpoint size is 418MB, the peak usage is 836MB.
>
>
> [1] Structured Pruning of Large Language Models https://arxiv.org/pdf/1910.04732.pdf
> [2] https://github.com/Arexh/BERT-Pruning

---

> > ### Comment · Reviewer_o4hz · 2021-08-22
> > **Thank you for the response**
> >
> > I want to thank the authors for their detailed response. I can understand the difference between the proposed method and PCN, but still, those two methods use similar ideas. Thus, it is worth at least discuss PCN.
> >
> > I do not think reporting BERT-based results is a standard in this field. I tend to believe the reason is most of the research groups do not have access to advanced computing resources. Thus, adding BERT-large/GPT results will definitely make the paper stronger.
> >
> > I understand that adopting DRONE to speed up pre-training is not trivial, but is it at least possible to speed up fine-tuning?

---

> > > ### Author Response · Authors · 2021-08-24
> > > **Additional Response**
> > >
> > > 1. We strongly agree with the reviewer that PCN is a related method and should be added into the discussion. We will discuss the difference between our method and PCN in the final version.
> > >
> > >
> > >
> > > 2. We thank the reviewer for requesting experiments on larger models! This will indeed make the paper stronger and we follow the request to demonstrate DRONE on BERT-LARGE with RTE dataset. We get a pretrained model from https://huggingface.co/yoshitomo-matsubara/bert-large-uncased-rte/tree/main and it achieves accuracy 74. BERT-LARGE doubled number of layers and the dimension is increased from 768 to 1024. Average inference time for a data increased to 1405ms. We perform DRONE and retrain. The overall result is 72.9 with inference time 1018ms (1.38x speedup). This result is comparable to our BERT-BASE result (1.31x speedup). Also, using the same rank with SVD + retrain can only achieve accuracy 56.2. Difference between SVD and DRONE is much more prominent on BERT-LARGE. Our preliminary result shows that DRONE can also work on larger models such as BERT-LARGE and we will add this discussion and results on more dataset in the final version.
> > >
> > >
> > >
> > >
> > > 3. We agree with the reviewer that fine-tuning is more applicable so we tried to apply DRONE in fine-tuning, again with RTE and we demonstrate it directly on bert-large models. Our setup is as follows. We get bert-large-uncased model from https://huggingface.co/bert-large-uncased/tree/main. We then perform DRONE on the model to obtain a low-rank structure, and then use the same fine-tuning parameter as the model in our reply 2 (https://huggingface.co/yoshitomo-matsubara/bert-large-uncased-rte/tree/main).
> > >
> > > We found out this procedure with directly using the same rank as in our reply 2 ( with 1.38x speedup) can only achieve 68.2 accuracy. But if we reduce the compression ratio into 1.2x training time speedup, this procedure can give us 72.9 accuracy. On the other hand, the same procedure with SVD as the initilialization of low-rank structure can only get 52.1 accuracy. This preliminary experiment shows that in addition to inference time acceleration (our original target), DRONE also has the potential to be applied in the training, but directly transport DRONE into training can not lead to the optimal result. There might be some issues needed to be studied such as learning rate schedule, batch normalization and number of training epochs in order to achieve the best results. We want to emphasize again that our work focuses on inference time acceleration, but the discussion of potential usage in training is interesting. We will add these results and discussions in the final version and this is certainly an interesting future direction.

---

> > > > ### Comment · Reviewer_o4hz · 2021-08-30
> > > > **Thank you for your response**
> > > >
> > > > After reading through the authors' responses. My main concerns are addressed. I will increase my overall evaluation score to 6.

---

### Official Review · Reviewer_Sxsk · 2021-07-14

**Rating:** 6
**Confidence:** 4

**Summary:**

This paper introduces DRONE, which is a data-driven low-rank decomposition method. Experiments are conducted using LSTM on PTB dataset and BERT on GLUE tasks. The method performs better compared to naive SVD.

**Limitations And Societal Impact:**

Yes

**Main Review:**

Strengths:
- The method is simple and orthogonal to other methods.
- Writing is clear and straightforward.
- Authors report the performance of DRONE combined with distillation and quantization methods, which demonstrate the potential for production.

Weaknesses & Questions:
- For the detailed time cost (of BERT), I only can find the numbers of DRONE, can you also provide the costs of baseline?
- How do you measure speed-up? What batch-size is used? Line 231-232 looks not very clear. I guess you use large batch-size for evaluation, then use the average time per sample? If yes, can you also provide the speed-up of single sample inference? (As single-sample inference time is important to real-world)
- In real-world products, we often need to re-train / re-finetuning the model by the latest data, due to distribution changes. As DRONE is data-depend, will distribution-change cause more performance loss?
- Did you try DRONE on language generation tasks? How does it perform?

**Time Spent Reviewing:**

8

---

> ### Author Response · Authors · 2021-08-10
> **Thank reviewer for valuable opinions and we will reply to each individual point**
>
> - On BERT-BASE time and number of samples to calculate time
>
> Different datasets differ slightly, but on average inference time of all tasks are within the same range. Take MRPC for example, Attention 131.6 ms, Attention layer FF0 29.9 ms, FF1 layer 140.4 ms, FF2 layer 127.6 ms, other operations 3.0ms. Total Inference: 432.5ms. We will add in the final version.
>
> On measuring the speedup, the reviewer’s understanding is correct. We use a batch size of 100 and report average. However, we use single-thread to measure the performance in order to exclude speedup due to the BLAS parallel operations. Thus, results of size 1 and our reported result don't differ a lot. For MRPC, it takes on average 433.2ms for a single data and 432.5 for the batch-averaged result. If we enable the system to use multi-threading BLAS, the average inference time shrinks to 52ms for batch calculation. We report single-thread setup as we want to present the most resource-limited environment such as mobile/edge devices.
>
> - On data distribution shift
>
> DRONE essentially tried to approximate the behavior of the original model under a generalized low-rank view. So if the data distribution changes but the original model’s performance is maintained, DRONE will not be affected. But if this data distribution shift causes the original model to fail then DRONE will also suffer since we are compressing the original model.
>
> - On language generation
>
> We thank the reviewer for bringing this up. We conducted the method on the machine translation task via OpenNMT. It provides a 2-layer transformer model on en-de translation. On the transformer part, DRONE achieves 1.76x speedup with BLEU from 33.47 to 33.26 which shows that DRONE also works on language generation tasks. We will add this experiment in our final version.

---

> > ### Comment · Reviewer_Sxsk · 2021-08-15
> > **speedup for single-data inference**
> >
> > Thanks for the responses.
> >
> > > Thus, results of size 1 and our reported result don't differ a lot. For MRPC, it takes on average 433.2ms for a single data and 432.5 for the batch-averaged result.
> >
> > Does it mean the speed-up is almost the same for single-data inference?

---

> > > ### Author Response · Authors · 2021-08-19
> > > **Speed up for single data**
> > >
> > > Yes for our results. This is true given that our result is reported under a single-thread program so that no parallelism can be used to accelerate the matrix computation.
> > >
> > >
> > > In a multi-threads (8) environment, batch result is 52 ms which is roughly 8x faster than single thread environment so in multi-thread environment batch-data can be sped up. Single-data under multi-threads (8) is still about 425ms, which does not appear to accelerate too much (compared to 432ms). Again, our results in the paper use a single thread so it makes almost no difference between single and batch data.

---

> > > > ### Comment · Reviewer_Sxsk · 2021-08-28
> > > > **thank you for the response**
> > > >
> > > > I will increase my score to 6.

---

### Official Review · Reviewer_i8zU · 2021-07-15

**Rating:** 6
**Confidence:** 3

**Summary:**

The paper proposed a data-aware low rank compression for BERT. It is stated that data-aware low rank compression could perform better than standard low rank compression like SVD since weights for matrix multiplications in BERT are not low-rank and SVD will fail. It seems that the motivation makes sense, and experiments evidenced its effectiveness.


**Limitations And Societal Impact:**

There is nothing related to Societal Impact. The Limitation is that the speedup is not notable while the performance is a little bit worse. This limits its applications in the real world. But the idea itself is intertesting.

**Main Review:**


It is not fair to say 12.3 times speed up in the Abstract, because  DRONE is used on TinyBERT but it seems that the speedup is calculated upon standard BERT. The speedup mainly comes from TinyBERT.

The example in Sec.3 (between the line 135 and 136) assumes that X is low rank. I am wondering whether this is true in real-world applications.

It seems that data-aware low rank compression is not specific to pre-trained language models. Is there any previous work that evidences its effectiveness in some simple scenarios like MLP or CNN, especially models are not needed to be pre-trained? By doing so, it may better investigate the behavior of  "data-aware low rank compression".

Can the authors show some information about the ranks {k} in different weights?

In Algo.2 both m_i and G are called 'module'. What is  G like? In the second for-loop, we find a subscript i in M and we also find i in the third for-loop.


In Table 2, why speedup in tinyBERT varies in different tasks? And RTE, why is tinyBERT only 1.8x times faster than BERT? but in other tasks, 10 more times speedup is achieved.

**Time Spent Reviewing:**

2.5 hours

---

> ### Author Response · Authors · 2021-08-10
> **We thank reviewer for the valuable opinions and we will reply in points.**
>
> - On speedup number in Abstract
>
> We agree with the reviewer and we will revise the words to fairly represent our contribution.
>
> - On if X is low-rank
>
> Basically we can observe this phoenema in all the dataset within our experiments. For example, PCA on X can explain 50% of the variance with only 25% ranks which is better than factorized weights (Figure 2., need 40% ranks to explain 50% variance). Despite the fact that we don’t have a theoretical guarantee on rank of X, notice that in eq(3), if X doesn’t represent significant low-rank structure, X will degenerate to some full-rank matrix and DRONE can still manage to solve the proposed optimization problem better than SVD.
>
> - On trying MLP/CNN
>
> Thank you for the suggestion. As presented in our paper, DRONE can work for any layer with dense matrix-matrix products, such as any MLP or LSTM (we have results for LSTM in Appendix D1). However, it may not be suitable for the convolution layer due to the convolution operator. Recently there are many convolution-free architectures in CNN proposed and DRONE should be able to apply to those networks as well.
>
> - On example of ranks
>
> We thank the reviewer for bringing this up. Just an example from SST-2 result. Below are ranks used for each module for each layer. We will add some of these examples in the revision.
> qkr = [192, 384, 192, 768, 768, 192, 768, 768, 192, 192, 768, 192]
> w0r = [288, 768, 96, 192, 288, 192, 768, 768, 96, 96, 288, 96]
> w1r = [96, 96, 768, 768, 288, 288, 768, 768, 96, 96, 288, 96]
> w2r = [192, 192, 768, 288, 768, 768, 768, 192, 96, 192, 96, 96]
> Basically, we can observe that the attention module is the hardest to be approximated and the top layers are easier to be compressed than the lower layers.
>
>
> - On example of G
> Sorry for the confusion. The most inner i should be a j to iterate through the set G, and each element in G should be noted as G_{l,i,j}( j-th hyperparameter for i-th module in l-th layer). G stands for  “grid” set of hyperparameters to search. Thus, for a module m_i in l-th layer, it has a corresponding sets of parameters G_{l.i} to search. It’s a subset of ranks to search. For example, if the full rank is 768, G_{l.i} can be a set as {48,96,192}. We will revise the algorithm in the final version.
>
>
> -On performance difference of RTE
>
> To set the hidden dimension of tinyBERT to r, it requires distilling a r-dimensional BERT model first, which takes days to train. We thus can only use author-released distilled tinyBERT models. However, there are only 2 models released and it happened that for RTE, the smaller one caused the accuracy drop significantly ( > 5%) so we can just compare the less distilled one with much conservative compression. This relates to the intrinsic difficulty of each task.

---

> > ### Comment · Reviewer_i8zU · 2021-08-11
> > **On if X is low-rank**
> >
> > Thanks for your response.
> >
> > **Regarding your reply about "On if X is low-rank"**
> >
> > Should X  be the (input) data, or weight matrices?

---

> > > ### Author Response · Authors · 2021-08-11
> > > **Regarding "On if X is low-rank"**
> > >
> > > In the reply, we refer X to be the distribution of data, which complies with our notation in eq (3). For the numbers, it's from the data distribution fed into the last FF-2 layer of fine-tuned SST-2 task.

---

> > > > ### Comment · Reviewer_i8zU · 2021-08-18
> > > > **if data distribution X is low-rank**
> > > >
> > > > Thanks for your clarification.
> > > >
> > > > Let us suppose each data point is represented in a D-dimensional feature vector, and there are N training examples. Do you mean the matrix (in size of D*N) is low-rank?  Or in a batch with the batch_size being B, the feature matrix in a batch (in size of B*D) is low-rank?
> > > >
> > > > It seems that figure 2 is about weight matrices, not data.
> > > >
> > > > Let me know if I was wrong.

---

> > > > > ### Author Response · Authors · 2021-08-19
> > > > > **PCA X**
> > > > >
> > > > > In our reply, “PCA on X can explain 50% of the variance with only 25% ranks which is better than factorized weights” refers to matrix X ( the size of DN matrix ) represents a “more low-rank” structure than  only exploring weight matrix. Since N can be large in many cases, in our experiment we subsample 10% of total training instances to approximate the distribution of X.It’s a property regarding the whole training dataset but not just a batch of data (size of BD).
> > > > >
> > > > > Figure 2 indeed is about weight matrices. But what it shows is that in general these weight matrices are “NOT” low-rank as it needs about 40% rank to explain 50% of the variance.

---

> > > > > > ### Comment · Reviewer_i8zU · 2021-08-24
> > > > > > **how you calculate PCA on these dataset ?**
> > > > > >
> > > > > > Sorry for my previous misunderstanding. Thanks for your clarification.
> > > > > >
> > > > > > The benchmark you are using is GLUE, each task of which involves text as the training example. Do you calculate PCA on the text itself? Text is composed of discrete tokens, I am curious how you exactly define the X when you report that “PCA on X can explain 50% of the variance with only 25% ranks”. Let me assume that PCA is calculated on the hidden states that represent each token of the training example, or a whole example. is it related to a  hidden state of each token, or hidden states of all tokens that make a training example?
> > > > > >
> > > > > > And does this observation (“PCA on X can explain 50% of the variance with only 25% ranks”) vary among different layers?  does this observation vary among different tasks (from SST-2 to MNLI)?
> > > > > >
> > > > > > I know that doing subsample is efficient, do you think the observation ((“PCA on X can explain 50% of the variance with only 25% ranks”) ) may still hold if you use full training examples?
> > > > > >
> > > > > > Best

---

> > > > > > > ### Author Response · Authors · 2021-08-24
> > > > > > > **calculate PCA on these dataset**
> > > > > > >
> > > > > > > 1. The example you describe is the input X to the first layer of the model, and your understanding is correct. The text is transformed into embedding and fed into the first layer. So for the first layer, "X" is referring to the embedding vectors of each text token. However, X cannot be interpreted as the token embedding in the upper layers. The attention modules and two feedforward layers will allow the model to learn a highly nonlinear interactions between tokens, so the output of the first layer will no longer be representing a simple word embedding vector. It's the embedding containing context information (from attention interaction) useful for the final prediction task.
> > > > > > >
> > > > > > > Thus, to answer your next question:
> > > > > > >
> > > > > > > "the hidden states that represent each token of the training example, or a whole example. is it related to a hidden state of each token, or hidden states of all tokens that make a training example?"
> > > > > > >
> > > > > > > It's more of the case "hidden states of all tokens that make a training example". We follow the standard setup in the huggingface pytorch models to use "sequence length" 128. In other words, the text will firstly be transformed into tokens and training tokens are dissected into many chunks and each chunk is of size 128 (padding 0 if necessary). Say we have total N chunks of text of 128 tokens, embedding size is 768. Total training text is a matrix of suze ( N * 128 ) x 768. During training, each time a batch size B will be sampled (out of N) and the data in shape B x 128 x 768 will be fed into the model. However, the PCA we perform to analyze is not on the batch but on all the N x 128 vectors each with size 768. Usually N is large so N x 128 is much larger than 768, and hence collected matrix (N*128) x 768 has max rank 768. We perform PCA on this to find out that about 192 ranks (25%) is enough to explain 50% variance.
> > > > > > >
> > > > > > >
> > > > > > >
> > > > > > > 2. On the question "And does this observation (“PCA on X can explain 50% of the variance with only 25% ranks”) vary among different layers? does this observation vary among different tasks (from SST-2 to MNLI)?"
> > > > > > >
> > > > > > > ANS:  In general yes, but the ratio differs between layers and datasets. Different tasks might exhibit different degree of low-dimensional property of data. In our first reply, we want to emphasize that yes there might be some differences across layers/datasets, but our proposed optimization problem can still work to find a better low-rank solution of the model. Even if X is totally not low-rank, the proposed DRONE can still work as the problem just degenerate back to the classical SVD so our solution is strictly better than common low-rank methods.
> > > > > > >
> > > > > > > 3. On the question "using full data rather than subsampling":
> > > > > > >
> > > > > > > Thanks for asking this question! In principle, more accurate distribution of X we have, the better approximation our eq(2) problem can provide. Thus, having more data is actually beneficial. We have experimented on smaller dataset such as MRPC, we found out that full-data will help a bit (from 88.0 to 88.2) but no significant improvement found when full-data is used. Thus, we apply subsampling to accelerate the DRONE, especially for large datasets. We will be adding this part in the final version to clarify the confusion.

---

> > > > > > > > ### Comment · Reviewer_i8zU · 2021-08-24
> > > > > > > > **Thanks for your reply**
> > > > > > > >
> > > > > > > > I appreciate your efforts to reply to my questions.  The idea itself is interesting. I increased the score from 5 to 6.

---

### Official Review · Reviewer_AXUP · 2021-07-19

**Rating:** 7
**Confidence:** 4

**Summary:**

This paper proposes DRONE, a data-aware low-rank compression algorithm for BERT, achieving a speed-up with marginal accuracy loss.

**Limitations And Societal Impact:**

Developing a better algorithm for the distribution of loss ratio would be an interesting research direction. Instead of equally distributing in a deterministic manner, dynamic programming based method can be used for optimal distribution.

Inspired by many works on iterative pruning, I am curious whether DRONE can be performed iteratively (e.g., DRONE → finetune → DRONE → ...) for better final performance despite the increased computational cost.

**Main Review:**

DRONE effectively reduces the computational cost with the observation that intermediate feature vectors lie on a low-dimensional manifold. DRONE significantly outperforms the vanilla SVD method.

I presume how to distribute r into each module R_{l, i} is important for the final performance. Algorithm 2 is duplicated in Section 3.3 and Appendix B. Although there are explanations about Algorithm 2, it would be better to describe its rationale. I think the method is somewhat heuristic. Do you have any results on other design choices? Small errors in a lower layer can be propagated to a higher layer resulting in a big difference to the final representation. Is there any consideration for this effect in the proposed method? Also, in this sense, I wonder whether DRONE is also effective in deeper networks (e.g., BERT-Large).

The combination of DRONE with other compression methods is practically valuable. The authors show that DRONE can be used orthogonally with knowledge distillation and quantization. DRONE could be compared with Linformer which also leverages low-rank structures.

Detailed analysis on ranks for each position (self-attention or feed-forward / layer) would give an insight for a future design of new architectures or compression methods.

Figure 2 only shows singular values at the very first and last layer. I am curious whether the same trends appear at intermediate layers. Considering a space limit, the authors may report the accumulation ratio of singular values choosing the top 50% rank, for instance.

Minor: It is strange that the body of the appendix is aligned to the center.

**Time Spent Reviewing:**

7 hours

---

> ### Author Response · Authors · 2021-08-10
> **Thank reviewer for valuable opinions and we will reply to each individual point**
>
> - On rationale of distributing R
>
> Basically we want to distribute according to its computational complexity. If a module requires longer computation time, we assume it's more complex and we want to distribute more allowed loss increase (budget) into the approximation. We agree with reviewer that exploring more disciplined distribution can be an interesting direction.
>
> - On loss propagation to higher layer
>
> We mitigate this effect by starting approximation on the lower layer first and then freeze their parameters when approximating upper layer parameters. This will minimize the potential effect of error propagation.
>
> - On larger models
>
> In the early stage of the project, we have tried DRONE on BERT-Large which achieves 1.5x speedup with < 1% accuracy loss. However, we observed that most of the existing works are comparing within BERT-base so later on we focus on the experimental results with BERT-BASE.  Essentially DRONE is a more general factorized matrix computation. Thus, as long as the underlying computation is matrix multiplication, DRONE should work but it takes longer time to complete the approximation for large models.
>
> - Detailed analysis on ranks for each position
>
> Take SST-2 for example. Below are ranks used for each module for each layer.
> qkr = [192, 384, 192, 768, 768, 192, 768, 768, 192, 192, 768, 192]
> w0r = [288, 768, 96, 192, 288, 192, 768, 768, 96, 96, 288, 96]
> w1r = [96, 96, 768, 768, 288, 288, 768, 768, 96, 96, 288, 96]
> w2r = [192, 192, 768, 288, 768, 768, 768, 192, 96, 192, 96, 96]
> Basically, we can observe that the attention module is the hardest to be approximated and the top layers are easier to be compressed than the lower layers.
>
> - On behaviour of intermediate layers
>
> It’s similar to top/bottom layers but can have different behaviour for different modules. Take MRPC for example, FF1 of layer 6 takes 35.1 % (r = 270) to achieve 50% explanation which is similar to top/bottom layers. But query/key matrices are more low-rank with 23% (r=175) to achieve 50%.
>
>  - On iterative DRONE and finetune
>
> Currently, we process all modules in a layer simultaneously. We do believe the best way is the iterative procedure as the reviewer suggested. We could process attention first -> fine-tune -> F0 layer -> fine-tune ->F1….-> F2-> fine-tune, which should give us better results but also requires longer fine-tuning time in total.

---

### Decision · Program_Chairs · 2021-09-27

**Decision:**

Accept (Poster)

**Comment:**

This paper proposes DRONE, a data-aware low-rank compression algorithm for BERT, achieving a significant speed-up with marginal accuracy loss. The authors show that DRONE can be used orthogonally with knowledge distillation and quantization. In this sense, the paper has its clear value to the community. The reviewers raised some concerns on the technical novelty, the precise quantification of the speed-up, and the experimental details. Overall speaking, the authors did a good job in their rebuttal, and some of the reviewers’ concerns were successfully addressed. The reviewers made several rounds of discussions, and the conclusion is that it would be good to accept the paper as a poster.